# IL-1 Superfamily Member (*IL-1A*, *IL-1B* and *IL-18*) Genetic Variants Influence Susceptibility and Clinical Course of Mediterranean Spotter Fever

**DOI:** 10.3390/biom12121892

**Published:** 2022-12-17

**Authors:** Letizia Scola, Giovanni Pilato, Rosa Maria Giarratana, Giuseppa Luisa Sanfilippo, Domenico Lio, Claudia Colomba, Giovanni Maurizio Giammanco

**Affiliations:** 1Clinical Pathology, Department of Bio-Medicine, Neuroscience, and Advanced Diagnostics, University of Palermo, 90133 Palermo, Italy; 2ICAR-CNR, Consiglio Nazionale delle Ricerche, 90146 Palermo, Italy; 3Microbiology, Department of Health Promotion, Mother and Child Care, Internal Medicine and Medical Specialties, University of Palermo, 90133 Palermo, Italy; 4Interdepartmental Research Center “Migrate”, University of Palermo, 90133 Palermo, Italy; 5Infectious Diseases, Department of Health Promotion, Mother and Child Care, Internal Medicine and Medical Specialties, University of Palermo, 90133 Palermo, Italy

**Keywords:** Mediterranean Spotted Fever, IL-1 super family, *IL-1* SNPs, genetic risk factors, decision tree methodology

## Abstract

Mediterranean Spotted Fever (MSF) is one of the most common spotted fever Rickettsioses. Most cases of MSF follow a benign course, with a minority of cases being fatal. The severity of the infection depends on bacterial virulence, dose and host factors such as effective immune response and genetic background. Herein, we reported data on typing by competitive allele-specific PCR of functionally relevant polymorphisms of genes coding for MyD88 adapter-like (*Mal/TIRAP*) protein (*rs8177374*), interleukin(IL)-1 cluster (*IL-1A rs1800587*, *IL-1B rs16944* and *rs1143634*) and *IL-18* (*rs187238*), which might be crucial for an efficient immune response. The results enlighten the role that *IL-1* gene cluster variants might play in susceptibility against *Rickettsia conorii* infection. In particular, the *IL-1A rs1800587TT* genotype was significantly increased in patients alone and combined in a haplotype composed by minor alleles *rs1800587T*, *rs16944A* and *rs1143634A*. This result was confirmed using the decision tree heuristic approach. Using this methodology, IL-1A rs1800587TT genotype was the better discrimination key among MSF patients and controls. In addition, the *IL-1* gene cluster SNP genotypes containing minor alleles and *IL-18 rs187238G* positive genotypes were found as associated with risk of severe complications such as sepsis, septic shock, acute respiratory distress syndrome and coma. In conclusion, these data suggest that the evaluation of *IL-1A*, *IL-1B* and *IL-18* gene SNPs can add useful information on the clinical course of patients affected by Mediterranean Spotted Fever, even if further confirmatory studies will be necessary.

## 1. Introduction

*Rickettsiae* are bacteria obligate to intracellular parasitism that cause potentially life-threatening illness worldwide, with fatality rates as high as 30% if not treated promptly [1] and are an important cause of emerging infectious diseases in people and animals. Rocky Mountain Spotted Fever caused by *Rickettsia rickettsii* and Mediterranean Spotted Fever (MSF) caused by *R. conorii* are the most common spotted fever Rickettsioses with a wide geographic distribution. MSF is typically characterized by fever, skin rash and black eschar at the site of the tick bite (tache noire) [2]. Most cases of MSF follow a benign course, with a case fatality rate of 3–7% among hospitalized patients [3,4]. The severity of *Rickettsial* infection is dependent on bacterial virulence, bacterial dose and in large part determined by host factors [5,6] and in particular by an effective immune response [7]. Resistance or susceptibility is strictly dependent on clearance of *Rickettsiae*. *Rickettsial* PAMPs (pathogen-associated molecular patterns), recognized by Toll Like receptor (TLR)4 and TLR2 on antigen presenting cells [8], and activating the cytotoxic pathway and T helper 1 responses [9,10]. Increased inflammatory responses characterized by Interferon (IFN)-α, Tumor Necrosis factor (TNF)-α, and Interleukin(IL)-1 increased serum levels have been reported in both human mild and severe *Rickettsia* infections [11]. The central role of the TLR4 in protection against *Rickettsia* infection in humans and in animal models has been described by different groups [10,12]. Actually, effective immune response against *Rickettsiae* is strictly dependent on the efficiency of first steps of immune response involving antigen presenting cells as dendritic cells (DCs). TLR4 activation signaling is initiated through MyD88 adapter-like (Mal) protein, encoded by the TIR Domain Containing Adaptor Protein (TIRAP) gene. The integrity of this pathway is essential for the activation of NFkB and inflammasome [13] and promoting transcription the of MHC class II and crucial pro-inflammatory cytokine and chemokine genes such as IL-1 and IL-18 [14]. Inflammasomes are intracytoplasmatic immune signaling complexes (categorized into canonical and non canonical) primed by exogenous PAMPs and their receptors that, in turn, lead the activation of *IL-1* and *IL-18* genes and the production and maturation of active cytokines [15]

IL-18 is a proinflammatory cytokine that was originally described as an interferon-gamma (IFN-γ)-inducing factor. Actually, IL-18 is considered a bridge among innate immune responses activated by Toll-like receptor (TLR) 2/4 and the myeloid differentiation primary response 88 (MyD88-dependent proinflammatory signaling pathway) [16]. Initiation of the canonical inflammasome results in the maturation of IL-1β, IL-18, and the activation of the pyroptosis mechanism. Although IL-1β is released by both canonical and non canonical inflammasome pathways, IL-1α is preferentially released by the non canonical inflammasome pathway, but other inflammasome-independent pathways might be involved [17,18,19,20]. Given the importance of both canonical and non canonical inflammasome-mediated IL-1 signaling in limiting pathogen colonization, many bacteria have evolved strategies to block their activation [21,22]. In particular, pathogen *Rickettsia* species activate evading mechanisms that involve the reduction of IL-1, in particular IL-1 α, production [23].

In this scenario, genetically determined protection or susceptibility factors might play a role. This is particularly evident in mouse models [24,25,26]. For example, C3H mice are genetically susceptible to a high, but not a low, dose of *R. conorii*, whereas C57BL/6 (B6) mice are highly resistant to both inocula.

In previous papers [12,27] we have reported that the reduction of TLR4 signaling and IFN-γ production associated with common polymorphisms of their genes represent susceptibility factors for Mediterranean Spotted Fever in Sicilian population. Herein we reported data on the typing of functionally-relevant polymorphisms of TIRAP genes coding for Mal protein [28], IL-1 gene cluster [29,30], and IL-18 [31] genes that might be crucial for tuning the efficiency of immune response against *Rickettsia* infection. *TIRAP rs8177374*, *IL-1A rs1800587*, *IL-1B rs16944* and *rs1143634*, and *IL-18 rs187238* SNPs were typed and analyzed with both formal statistic tools and applying a software for machine learning driven data analysis based on the decision trees protocol. This approach generated a visually described algorithm (decision tree) that can be easily interpreted. A decision tree is a non-parametric supervised approach used mainly for classification tasks. The model exploits a set of labeled data to find a decision path that leads a given data pattern to be assigned to a given class. Decision trees are composed of decision nodes, i.e., the conditions to be verified, and leaf nodes, where the decision of the class to use for labeling the pattern is generated. Decision Tree Software’s task is to find a possible small and accurate tree that provides a reliable outcome. The learning process of a decision tree software starts from a known (labeled) dataset to decide the sequence of attributes to choose for partitioning unclassified data in a specific class (e.g., recognition of a subject as an MSF patient or healthy control). The goal is to reduce the impurity or uncertainty in the split data as much as possible. A subset of data is said to be pure if all the items in the subset belong to the same class [32,33]. We applied this methodology to the SNP typing dataset of patients and controls to test the possibility to discriminate among affected or not-affected subjects starting from their genetic characteristics.

## 2. Materials and Methods

### 2.1. Subjects

A total of 366 subjects of western Sicilian ancestry were recruited for this study. Blood samples of 160 patients affected by MSF were collected at Infectious Diseases Unit of the University Hospital of Palermo. Patients had characteristic signs and symptoms of active MSF (presence of fever, eschar at the site of tick bite, maculopapular rash). Serological conversion (high increasing levels of anti-*R. conorii* antibodies) and RT-PCR confirmed the diagnosis. All patients were recruited excluding the presence of other chronic diseases as immune disorders, diabetes, cardiovascular disease and cancer, as well as chronic alcoholism and glucose- 6-phosphate dehydrogenase deficiency. Two hundred and six seronegative healthy subjects matched for age, sex and geographic origin were enrolled as a control group. Demographic and clinical characteristics of patients and control subjects are summarized in Table 1. Our study was performed according to ethical standards of the Helsinki Declaration of the World Medical Association and Italian legislation, and was approved by the local institutional review board (Comitato Etico Palermo 1, protocol code CET1 01/2019, date of approval 16 January 2019). All participants gave their informed consent. Data were encoded to ensure privacy protection of patients and controls. Blood specimens from control subjects and patients were collected at Day 1 of hospital admission in tripotassium EDTA sterile tubes, stored at −80 °C, and then used for DNA extraction. All laboratory procedures were performed without knowledge about the sources of the biological specimens.

### 2.2. Laboratory Diagnosis of Rickettsia conorii Infection

Patient serological tests were performed at admission, during the acute phase, and in the convalescent phase, while molecular tests have been performed on admission only. *Rickettsia conorii* VIRCLIA^®^ IgM monotest and VIRCLIA^®^ IgG indirect chemioluminescent (CLIA) monotests (Vircell, Granada, Spain) were used for serological tests. VIRCLIA IgM or IgG monotest use the monotest format (a device including Calibrator, Diluent Conjugate and Substrate Chemiluminescence immunoassay analyzed onon an automatic VIRCLIA^®^ LOTUS SYSTEM (Vircell, Granada, Spain). A commercial real time polymerase chain reaction (RT-PCR) Kit was used for qualitative detection of *Rickettsia conorii* genome (Rickettsia conorii Real-TM, Sacace Biotechnologies s.r.l., Como, Italy). Briefly *Rickettsia conorii* DNA extracted from the specimens (see below) was amplified using Real-Time amplification and detected by fluorescent reporter dyes linked to hydrolysis probes specific for *Rickettsia conorii* DNA and Internal Control.

### 2.3. DNA Extraction

The whole peripheral blood samples, made incohable with EDTA, were frozen at −20 °C to optimize red blood cell lysis. After defrosting, the DNA was extracted using the “Magna Pure” 24 System automated extraction method (Roche Diagnostics S.p.A., Monza (MB), Italy). This approach, based on a solid-phase extraction method (capture of magnetic beads covered with high-DNA-affinity silica followed by DNA elution) allows an efficient and high DNA yield.

### 2.4. SNPs Molecular Typing

As reported in Table 2, we selected six functional and common SNPs of *IL-1 cluster*, *IL-6*, *IL-18*, *and of TIRAP* genes. dbSNP NCBI and the ENSEMBL database (http://www.ensembl.org/index.html, last access on 8 September 2022) were queried for the selection of SNPs. DNA samples were typed using dedicated and pre-made competitive allele-specific PCR (Polymerase Chain Reaction) assays (KASPar), based on Homogeneous Fluorescence Resonance Energy Transfer (FRET) detection, developed by K-Bioscience (K-Bioscience Ltd. Hoddesdon, UK) as previously described [34]. Genotypes were determined using the 7300 system SDS software, versus 1.3 (Applera Italia, MONZA (MB), Italy) sample by sample, on the basis of the detection of fluorescence signals (unique for homozygous samples, double for heterozygous samples).

### 2.5. Statistics

Allele and genotype frequencies were evaluated by gene count, using an online statistical analysis tool applied to the evaluation of SNPs (https://www.snpstats.net/start.htm, last access on 30 September 2022). Data were tested for goodness of fit between observed and expected genotype frequencies according to Hardy-Weinberg equilibrium, by Pearson’s distribution and χ2 tests. Significant differences in allele, homozygous and heterozygous genotype distributions among groups were calculated by using Fisher’s exact test (adjusted by age and sex). Multiple logistic regression models were applied using dominant (major allele homozygotes versus heterozygotes plus minor allele homozygotes) and recessive (major allele homozygotes plus heterozygotes versus minor allele homozygotes) models. Odds ratios (OR), 95% confidence intervals (95% C.I.) and *p* values (*p*-value cutoff < 0.05) were determined using GraphPad InStat software version 3.06 (GraphPad, San Diego, CA, USA) and the abovementioned online statistical analysis tool. Haplotype frequency estimation (frequency threshold for rare haplotype: 0.01) and haplotype association with response estimation functions at https://www.snpstats.net/start.htm (last access on 30 September 2022) were applied to search for comparison of haplotype frequencies between patient and control groups at the *IL-1* gene cluster.

### 2.6. Decision Tree Model

Classification and Regression Trees (CART) implementation has been used [32] to check the possibility to built a computer-driven decision process that might be useful in discriminating MSF affected or not-affected subjects starting from the genetic background variations. The methodology builds the tree by exploiting the attribute and the decision threshold that yield the largest information gain at each node from time to time. Two common heuristics used are the Gini Index and the Entropy measure (Decision Trees). The dataset is composed of 366 entry containing genetic data and the classification label. We split 80% (292 random items) for training the tree and the remaining 20% (74 random items) for testing the tree. The training and the test dataset have the same distribution of cases and controls. The Gini index and the Entropy measure were applied to induce the model from the data using the Decision Tree algorithms at https://scikit-learn.org/stable/modules/tree.html (last access 10 October 2022). The results of this approach were analyzed for sensitivity, specificity and positive and negative predicting values [37,38].

## 3. Results

### 3.1. Analyses of Association of SNP Alleles and Genotypes to Mediterranean Spotted Fever Susceptibility

Analyses of SNP frequencies in 160 MSF patients and 206 healthy controls, after correction by age and sex, demonstrated that the *IL-1A rs1800587T* allele was significantly increased in patients with a strongly increased frequency of TT genotype associated with an increased production of IL-1α [29]. No significant differences were observed analyzing allele or genotype frequencies of the other SNPs (Table 3).

Considering that IL-1 gene cluster lays on a relatively small portion of chromosome (Chr) 2, using the Haplotype frequency estimation (frequency threshold for rare haplotype: 0.01) and Haplotype association with response estimation functions at https://www.snpstats.net/start.htm, (last access on 30 September 2022) eight *IL-1A rs1800587/IL-1Brs16944/rs1143634* haplotypes were identified. As reported in Table 4, the frequency of the haplotype number six, composed by minor alleles *rs1800587T*, *rs16944A* and *rs1143634A*, all known as markers of a higher production of both IL-1A and IL-1B, was significantly increased in the patient group.

### 3.2. Application of a Decision Tree Model Based on Genetic Background Variations for the Characterization of Mediterranean Spotted Fever Subjects

Considering that genetic markers might be useful to refine diagnoses, we applied a decision tree methodology (see materials and methods) to our results.

A Decision Tree algorithm, using SNP typings from 292 random selected cases and controls for training and 74 “unknown” for testing, identified the presence of IL-1A rs1800587TT genotype, accompained or not by homozygous genotypes of TIRAP rs8177374, as the label for MSF patient status and the absence as the label of healthy control status with an accuracy of 0.6486 (Figure 1). In other terms, this means that the probability that a subject bearing, or not, the IL-1A rs1800587TT genotype is assigned correctly to the MSF or healthy status applying the thi decision tree model about 65% of the time. Performances of the test are reported in Table 5. Eight of the 32 MSF cases used as “blinded” test resulted True Positive (MSF patients bearing rs1800587TT genotype) and 24 false negative (MSF patients negative for rs1800587TT genotype) whereas only 2 out 42 controls used as test were false positives. Accordingly, the sensitivity of the test was low, whereas the specificity and positive predicting values were, respectively, 95.2% and 80%.

### 3.3. SNP Genotype Association to Clinical Severity of Mediterranean Spotted Fever Symptoms

Since the genetic background, in addition to influencing the susceptibility or, on the contrary, protection towards an infectious disease, can also have a role in modulating the severity of the infection, we evaluated the association of the SNPs typed for the genes encoding TIRAP, IL-18, IL-1α and IL-1β proteins with the presence of MSF complications. The complete analyses of data stratified according to the presence of complicated MSF for all the five SNP typed are reported in Appendix A (Analysis of SNPs association with Acute Respiratory Distress Syndrome), Appendix A (Analysis of SNPs association with Neurologic symptoms), Appendix A (Analysis of SNPs association with with Sepsis), Appendix A (Analysis of SNPs association with Septic shock) and Appendix A (Analysis of SNPs association with Coma). Significant results were obtained for *IL-18 rs187238* SNP (Table 6) and for SNPs located in the *IL-1* gene cluster (Table 7). The data reported in Table 6 indicated that the presence of the G allele at *IL-18 rs187238* associated with an increased production of the cytokine [31] seems to be associated with an increased risk of ARDS in patients affected by MSF. On the other hand, our results, obtained on 15 MSF patients affected by ARDS, do not allow identifying *rs187238GG* homozygous individuals in this group, so this result should be considered with caution.

As reported in Table 7, the analysis of the genotypic frequencies of the group of patients who presented a more severe course with the presence of complications compared with both patients without complications and compared with controls, allowed us to identify the association of the presence of sepsis, septic shock, neurological syndromes and coma with some SNP genotypes located in the *IL-1* gene cluster. In particular, the presence of the homozygous *IL-1A rs1800587TT* genotype appears to be a significant risk factor for sepsis, septic shock and neurological symptoms. Similar results were obtained by analyzing the association of allele A even if in heterozygosity of SNP *IL-1B rs16944* and *IL-1B rs1143634TT* genotype. On the other hand, only 33 patients (Table 1) had one or more serious complications of MSF, so the data reported in Table 6 (21 patients with Neurological Symptoms, 17 patients with sepsis, 15 patients with Septic shock and 8 patients with Coma) must be considered preliminary and it is mandatory that they be confirmed in a much larger sample of patients. However, these results seem to indicate that the presence of genotypes containing the minor alleles of the SNPs *rs1800587*, *rs16944* and *rs1143634* may be a risk factor for the appearance of the most severe clinical pictures of MSF.

## 4. Discussion

Immune response against *Rickettsia* infections involves an adaptive and innate immune response. Studies on patients or mouse models emphasize the role of pro-inflammatory cytokines [7]. In addition, several members of the Toll-like receptor (TLR) family are involved in host response to *Rickettsiae*, and the integrity of their signal transduction pathway appears to be crucial for an effective immune response [39]. In an experimental model, both *Rickettsia conorii* and *Rickettsia australis* (the etiologic agent of Queensland tick typhus) infections are favored in MyD88 knockout mice. Furthermore, the secretion levels of IL-1β by Rickettsia-infected mice were significantly reduced in MyD88−/− mice compared with WT controls, suggesting that in vitro and in vivo production of IL-1β is dependent by on integrity of TLR transduction signals [39].

At the intra-cytoplasm level, the TLR signaling pathway activates canonical and non-canonical inflammasome pathways that survey the invading pathogen’s inflammasome and subsequently trigger maturation (activation) of pro-forms of the IL-1 cytokine family components as IL-18 (pyroptosis) [18,26]. Activated IL-1 and IL-18, as proinflammatory cytokines, are involved in the restriction of bacterial replication by limiting the replication niche of intracellular bacteria, as demonstrated for *R. australis* infection in a mouse model and in the human disease [40].

In this view, we evaluated some functionally relevant SNPs that might modify the activation of inflammasome (in particular *TIRAP rs8177374*) and related cytokine production (*rs1800587*, *rs16944* and *rs1143634* of *IL-1 gene cluster* and *IL-18 rs187238*). *TIRAP rs8177374* is a non-synonymous SNP encoding a thymine (T) instead of a cytosine (C) with the transcription of a MAL protein with a leucine instead of a serine at position 180 (S180L). Cristal structure studies [35] of *TIRAP* gene have demonstrated that S180 residue is located in a surface-exposed cavity and that substitution by leucine is likely to cause steric occlusion of the cavity. MyD88 may bind in a different configuration to the MAL S180L variant, which interferes with the formation of the receptor ternary complexes required to initiate signaling [35].

In some infectious disease models as malaria, the presence of CC genotype results in excessive production of inflammatory cytokines; the TT genotype results in a compromised immune response and CT genotype carriers result in the production of a balanced immune response [28,41]. Our data indicate that *rs8177374* is not associated with susceptibility or protection against MSF, in spite of our previous results demonstrating that TLR4 genetically-determined signaling attenuation represents a susceptibility factor for Mediterranean Spotted Fever in Sicilian population [12]. Similar results were obtained for *IL-18 rs187238* SNP.

On the other hand, data herein reported emphasize the role that *IL-1* gene cluster variant combinations might play a role in susceptibility or protection against *Rickettsia conorii* infection. Actually, our data indicated that *IL-1A rs1800587TT* genotype frequency, associated with a higher production of the cytokine [29], was significantly increased in patients. Furthermore, when the association of the *IL-1 gene cluster* haplotypes were analyzed, a haplotype composed by minor alleles *rs1800587T*, *rs16944A* and *rs1143634A*, all known as markers of a higher production of both IL-1α and IL-1β cytokine [36], was found significantly increased in the patient group with respect to the control group. These data apparently are in contrast with findings that indicate that a winning strategy of pathogen *Rickettia* spp. (as *R. typhi* and *R. rickettsia*) to overlay intracellular host defense is the inhibition of inflammasome-activated IL-1 production [23]. However, in different infection models, e.g., in SARS-CoV 2 infection, microbial components that, in the first phase, reduced the expression and production of IL-1 and IL-18, interfering with inflammasome-activation mechanisms, in the later stages of infection, activated the same mechanisms enhancing inflammasome responses [42]. Actually, circulating proinflammatory cytokine and in particular IL-1 and IL-6 are increased as a consequence of *Rickettsia*-macrophage interactions [43].

As is well known, IL-1α is an alarmin released by dying cells, as Rickettsia infected endothelial cells are, to initiate the early phase of sterile inflammation, while IL-1β is produced by inflammasomes at sites of tissue infection or sterile injury [44]. These cytokines therefore have overlapping functions and the different activation pathways assure a fundamental mechanism of innate and adaptive immune response. Both IL-1α and IL-1β induce local inflammation and the recruitment and activation of neutrophils, monocytes and macrophages, which might act as a double-edged sword, limiting on infection the one hand and activating inflammation-related local and systemic damage on the other [44]. Alternatively, IL-1α, IL-1β, and IL-18 drive pathology in a range of human diseases, such as gouty arthritis, autoinflammatory disease [44], or cytokine storm in systematic infectious diseases [42]. The data reported herein seem to suggest that a genetically-determined high production of these cytokines changes the balance towards negative effects of IL-1 cytokine activation, facilitating MSF development after *R. conorii* infection and probably favoring a more severe clinical course. In this view, one can hypothesize that the typing of the IL-1 gene cluster might drive the use of innovative drugs, such as inflammasome modulators, to avoid severe complications in MSF and in other tick-borne related diseases.

To further explore the association of typed SNPs with MSF, we applied a heuristic methodology to build a decision tree that we tested for the ability to discriminate among MSF affected or not-affected subjects. Using this approach, we found that the *IL-1A rs1800587TT* genotype was the discriminating key to distinguish among MSF patients and healthy subjects with an accuracy of about 65% (Figure 1). All together, these results strongly suggest that the *IL-1A rs1800587TT* genotype plays a role in MSF susceptibility and opens the perspective of a diagnostic application of this approach. In theory, a patient with clinical pictures compatible with MSF might be tentatively classified as infected or not by a pathogen *Rickettsia* with a probability of about 65% after *IL-1A rs1800587* typing. However, even if the specificity and positive predicting value of the test was high in our experimental condition (cohort study), considering the relatively low prevalence of the disease in general population, a larger set of SNPs, probably combined with other clinical and laboratory markers, appear necessary to build an efficient diagnostic tool [38].

Then, we checked if these genetic markers play a role in modulating the severity of the disease, evaluating the association of the typed SNPs with the presence of severe complications of the disease. As reported in Table 1, thirty-three MSF patients experienced one or more severe complications such as sepsis and its sequelae, neurological symptoms (confusion, sleepiness or, in one case, seizure), coma or ARDS.

The analysis of the SNP genotypic frequencies in this group of patients compared with both patients without complications and controls shows that *IL-18 rs187238G* positive genotypes seem to be associated with an increased risk of ARDS in patients affected by MSF even if the small number of subjects affected by ARDS imposes to consider these data preliminary. *IL-18 rs187238G* allele positive variants are both associated with increased expression and/or production of the cytokines [31]. The G-to-C substitution at position -137 (*rs187238*) of *IL-18* gene insert a histone 4 transcription factor-1 (H4TF-1) nuclear factor-binding site. Functional assays have previously demonstrated that the *rs187238G* allele is associated with increased production of IL-18 [31,45] and correlates with increased IL-18 levels in peripheral blood mononuclear cells or plasma [46]. In this view might be of a certain interest that IL-18 levels are considered useful markers of ARDS in COVID-19 and other clinical conditions [47,48,49].

In addition, we found that genotypes of the *IL-1* gene cluster SNPs are associated with a higher production of the relative cytokines; in particular *IL-1A rs1800587TT* might be significant a risk factor for sepsis, septic shock and neurological symptoms. Even if these results appear coherent with those reported above, in delineating a picture in which positivity for genotypes containing the minor alleles of the SNPs *rs1800587*, *rs16944* and *rs1143634* of the *IL-1* gene cluster may be considered as susceptibility markers for MSF and probable indicators of severe complication risks, they should be considered as preliminary data, which should be confirmed in a much larger sample of patients.

## 5. Conclusions

The presence of the genotypes of SNPs associated with an increased production of proinflammatory cytokines such as IL-1α, IL-1β and IL-18 can constitute susceptibility factors for the development of Mediterranean Spotted Fever and for the development of severe, potentially life-threatening complications. While further confirmatory studies are needed, the data presented in this paper suggest that the evaluation of these and other SNPs of genes encoding IL-1α, IL-1β and IL-18 [36] can serve as useful diagnostic and prognostic information in the clinical course of patients affected by Mediterranean Spotted Fever that can drive innovative therapeutic approaches.

## Figures and Tables

**Figure 1 biomolecules-12-01892-f001:**
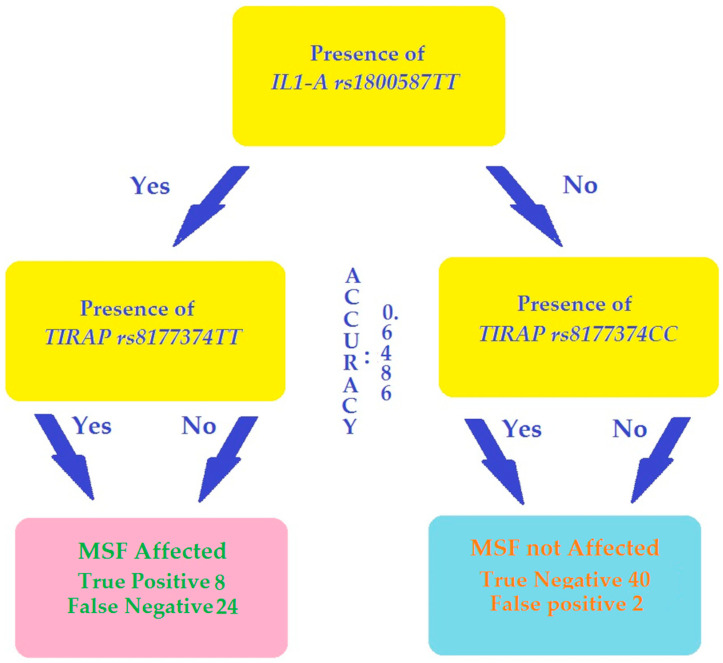
Decision tree algorithm. Data from 292 random selected case (128) and control (164) SNPs typing were used as a training set allow elaborating a decision rule that associates the positivity for one or more SNP genotypes to the MSF-affected and negativity to MSF non-affected label. The better combined decision and leaf nodes generate a decision tree that assign the label using positivity or negativity for the IL-1A rs1800587TT genotype, in the presence or not of TIRAP rs8177374 homozygous genotypes with an accuracy of 0.6486 (accuracy range for the procedure from 0 to 1). Applying the decision tree to the 74 test subjects used as “blinded tests”, only eight of the 32 MSF cases were correctly recognized as MSF patients. On the other hand, 40 out 42 controls used as tests were recognized as healthy.

**Table 1 biomolecules-12-01892-t001:** Demographic, microbiological and clinical characteristics of 160 patients affected by Mediterranean Spotted Fever (MSF) and 206 control subjects.

Demographic Characteristics	MSF		Controls		*p*
	N.	%	N.	%	
Adult females	60	37.50	85	41.26	
Age, mean ± SD	41.84 ± 14.28		38.95 ± 12.61		0.863
Age median	39.00		39.50		
Clinical characteristics of Mediterranean Spotted Fever patients					
Fever Days ± SD	median		Fever T° Max ± SD		median
9.12 ± 4.86	8.00		39.39 ± 0.64		39.45
MSF Symptoms	N.		%		
Rash	151		94.37		
Tache noire	88		55.00		
Lymphadenitis	17		10.63		
Generalized lymphadenitis	7		4.38		
Splenomegaly	45		28.13		
Hepatomegaly	61		38.12		
Arthralgia	99		61.88		
Conjunctivitis	11		6.88		
Severe Complications					
All complicated cases	33		20.63		
ARDS *	15		9.38		
Sepsis	17		10.62		
Septic shock	15		9.38		
Neurologic symptoms	21		13.12		
Coma	8		5.00		

Number (N.) and percentage (%) have been reported, * ARDS: Acute Respiratory Distress Syndrome.

**Table 2 biomolecules-12-01892-t002:** Genes, reference number (rs), localization and position of SNPs investigated in the study.

Genes	SNPs	Gene Region	Position	Minor Allele	Biological Effect	References
*TIRAP*	*rs8177374*	Exon 5 +539	11:126,292,948	T	Reduced receptor signal transduction	[28,35]
*IL-18*	*rs187238*	5′ flanking region *−137*	11:112,164,265	G	Increased levels of gene transcription	[31]
*IL-1A*	*rs1800587*	5′ flanking region *−889*	2:112,785,383	T	Marker of Increased levels of gene transcription	[29]
*IL-1B*	*rs16944*	5′ flanking region *−511*	2:112,837,290	A	Marker of increased Active IL-1β release	[30]
	*rs1143634*	Exon 5 +3954	2:112,832,813	T	Marker of increased Active IL-1β release	[36]

**Table 3 biomolecules-12-01892-t003:** Analysis of allele and genotype frequencies (Freq.) of *TIRAP rs8177374C/T*, *TNF rs1800629*, *IL-18 rs187238G/C*, *IL-1A rs1800587C/T*, *IL-1B rs16944A/G* and *IL-1B rs1143634 C/T* polymorphisms in 160 patients affected by Mediterranean Spot Fever (MSF) compared with 206 healthy controls (adjusted by age and sex).

GENE	SNP	Alleles/Genotypes	Controls		MSF		OR (95% CI)	*p*-Value
			N.	Freq.	N.	Freq.		
TIRAP	rs8177374	C	378	0.92	291	0.91	1.11 (0.66–1.86)	0.639
		T	34	0.08	29	0.09		
		C/C	178	0.86	134	0.84	0.81 (0.45–1.45)	0.553
		C/T	22	0.11	23	0.14	1.39 (0.74–2.60)	0.480
		T/T	6	0.03	3	0.02	0.64 (0.16–2.59)	0.521
IL-18	rs187238	C	288	0.7	231	0.72	0.89 (0.65–1.44)	0.512
		G	124	0.3	89	0.28		
		C/C	102	0.5	82	0.51	1.07 (0.71–1.62)	0.750
		C/G	84	0.41	67	0.42	0.99 (0.64–1.53)	0.629
		G/G	20	0.1	11	0.07	0.63 (0.28–1.43)	0.333
IL-1A	rs1800587	C	287	0.7	196	0.61	1.45 (1.07–1.98)	0.018
		T	125	0.3	124	0.39		
		C/C	100	0.49	69	0.43	0.81 (0.53–1.22)	0.342
		C/T	87	0.42	58	0.36	0.97 (0.61–1.52)	0.241
		T/T	19	0.09	33	0.21	2.56 (1.39–4.70)	0.002
IL-1B	rs16944	G	272	0.66	202	0.63	1.14 (0.84–1.54)	0.436
		A	140	0.34	118	0.37		
		G/G	87	0.42	65	0.41	0.94 80.62–1.42)	0.831
		A/G	98	0.48	72	0.45	0.90 (0.60–1.36)	0.625
		A/A	21	0.1	23	0.14	1.48 (0.79–2.78)	0.222
	rs1143634	C	297	0.72	220	0.69	1.17 (0.85–1.62)	0.328
		T	115	0.28	100	0.31		
		C/C	108	0.52	81	0.51	0.93 (0.62–1.41)	0.753
		C/T	81	0.39	58	0.36	0.88 (0.56–1.36)	0.550
		T/T	17	0.08	21	0.13	1.77 (0.89–3.55)	0.134

Number (N.), frequency (Freq.), odds ratio (OR) and 95% Confidence Interval (95% CI) are reported.

**Table 4 biomolecules-12-01892-t004:** Analysis of eight *IL-1* gene cluster haplotype (Hp) frequencies (Freq.) in Mediterranean spotted Fever patients (MSF) and healthy controls (CTRL) (adjusted by age and sex).

Hp	*IL-1A rs1800587 C/T*	*IL-1B* *rs16944 G/A*	*IL-1B* *rs1143634 C/T*	CTRL Freq.	MSF Freq.	OR (95%CI)	*p*
1	C	G	C	0.3575	0.3204	0.84 (0.62–1.15)	0.31
2	C	A	C	0.2598	0.2051	0.74 (0.54–1.59)	0.096
3	T	G	T	0.1621	0.1444	0.86 (0.57–1.20)	0.54
4	T	G	C	0.0761	0.1071	1.49 (0.89–2.48)	0.15
5	C	G	T	0.0645	0.0593	0.92 (0.50–1.68)	0.88
6	T	A	T	0.0377	0.0811	2.19 (1.15–4.16)	0.016
7	T	A	C	0.0276	0.0549	1.81 (0.47–6.97)	0.055
8	C	A	T	0.0147	0.0277	1.96 (0.69–5.56)	0.29

**Table 5 biomolecules-12-01892-t005:** Performances of a decision tree algorithm obtained using SNP typings from 292 random selected cases and controls for training and 74 for testing.

Test Results	MSF Patients	Control Subjects	Total
Positive	8	2	10
negative	24	40	64
Total	32	42	74
Sensitivity	0.250		
Specificity	0.952		
PPV	0.800		
NPV	0.625		

PPV: positive predicting value; NPV: negative predicting value.

**Table 6 biomolecules-12-01892-t006:** Association of *IL-18 rs187238* genotypes with Acute Respiratory Distress Syndrome (ARDS) in 15 patients affected by Mediterranean Spotted Fever (MSF) compared to MSF without ARDS (w/oARDS) and Healthy Controls (adjusted by 64 years Age cut off and Sex).

Genes and SNP Alleles		ARDS		w/oARDS		Controls		ARDS vs. w/oARDS		ARDS vs. Controls	
		N.	Freq.	N.	Freq.	N.	Freq.	OR 95% CI	*p* Value	OR (95% CI)	*p* Value
*IL-18 rs187238*	C/C	2	0.13	80	0.55	102	0.5	0.12 0.03–0.57	0.0021	0.16 0.03–0.71	0.0069
	G/C	13	0.87	54	0.37	84	0.41	12.86 2.68–61.70	<0.0001	9.28 2.01–42.81	0.0005
	G/G	0	-	11	0.08	20	0.1	--	-	--	-

Number (N.), frequency (Freq.), odds ratio (OR) and 95% Confidence Interval (95% CI) are reported. Dominant model (GC/GG vs. CC) ARDS vs. w/o ARDS OR (95% CI): 10.99 (2.29–52.69) *p* < 0.0003; ARDS vs. Controls OR (95% CI): 6.23 (1.35–28.74) *p* = 0.0054.

**Table 7 biomolecules-12-01892-t007:** Association of *IL-1A rs1800587*, *IL-1B rs16944* and *IL-1B rs1143634* genotypes (odds ratio: OR, 95% confidence interval: 95% CI, significance: *p*) with the presence of neurological symptoms (Neu-Sympt, 21 patients), sepsis (17 patients), Sepsis shock (Sept-Sh, 15 patients) or coma (8 patients) in the complicated MSF patient group compared with the uncomplicated patient group and controls.

Genes and SNP Alleles		*IL-1A rs1800587*			*IL-1B rs16944*			*IL-1B rs1143634*		
		C/C	C/T	T/T	G/G	G/A	A/A	C/C	C/T	T/T
Sepsis vs. No-Sepsis	OR 95% CI	0.51 0.17–1.54	0.86 0.29–2.50	2.75 0.90–8.42	0.28 0.08–1.01	7.04 1.92–25.8	--	1.45 0.52–4.02	0.21 0.05–0.96	3.42 1.04–11.3
	*p* value	0.303	0.784	0.0871	0.0652	<0.0007	-	0.609	0.019	0.056
Vs. Control	OR (95% CI)	0.44 0.15–1.30	0.75 0.26–2.14	6.97 2.09–23.2	0.29 0.08–1.05	4.65 1.28–16.9	--	1.30 0.50–3.54	0.21 0.05–0.96	4.75 1.43–15.8
	*p* value	0.205	0.592	0.0027	0.069	0.0087	-	0.801	0.018	0.018
Sept-Sh vs. No-Sept-Sh	OR 95% CI	0.30 0.08–1.11	1.05 0.35–3.18	3.65 1.13–11.8	0.20 0.04–0.92	9.86 2.12–45.7	--	1.13 0.3–3.27	0.25 0.05–1.08	4.27 1.25–14.6
	*p* value	0.098	0.934	0.036	0.027	<0.0004	--	1.00	0.087	0.029
vs. Controls	OR (95% CI)	0.26 0.07–0.97	0.92 0.31–2.74	9.25 2.59–33.0	0.21 0.05–0.96	6.29 1.37–29.0	--	1.04 0.36–2.97	0.24 0.05–1.13	5.84 1.68–18.1
	*p* value	0.0351	0.881	0.001	0.034	0.0051	--	1.00	0.054	0.0097
N-Sympt vs. No-N-Sympt	OR 95% CI	0.62 0.24–1.63	0.79 0.29–2.11	2.47 0.87–6.96	0.21 0.06–0.73	9.83 2.74–35.3	--	1.35 0.54–3.42	0.39 0.12–1.22	2.40 0.76–7.58
	*p* value	0.357	0.634	0.0981	0.0085	<0.0001	--	0.641	0.083	0.164
N-Sympt vs. Controls	OR 95% CI	0.53 0.21–1.37	0.68 0.26–1.79	6.03 2.00–18.2	0.23 0.07–0.79	6.03 1.71–21.3	--	1.21 0.49–2.99	0.36 0.12–1.13	3.56 1.11–11.4
	*p* value	0.252	0.433	0.0024	0.0173	0.0011	--	0.819	0.059	0.046
Coma vs. No-Coma	OR 95% CI	2.99 0.68–13.3	2.68 0.58–12.3	0.19 0.02–1.64	9.52 1.14–79.7	--	1.67 0.38–7.22	0.25 0.03–2.15	2.24 0.41–12.1	2.99 0.68–13.3
	*p* value	0.143	0.225	0.1441	0.0096	--	0.720	0.151	0.384	0.147
vs. Controls	OR (95% CI)	2.36 0.54–10.28	9.00 1.75–46.30	0.195 0.02–1.62	6.90 0.83–57.6	--	1.51 0.35–6.49	0.23 0.03–1.91	4.06 0.73–22.48	2.36 0.54–10.28
	*p* value	0.241	0.015	0.145	0.033	--	0.725	0.111	0.154	0.242

Odds ratio (OR) and 95% Confidence Interval (95% CI) are reported; *IL-1A rs1800587*: Sept-Sh vs. Controls (Dominant model G/C-C/C vs. G/G) OR (95% CI): 4.23 (1.13–15.85), *p* = 0.018. *IL-1B rs16944*: N-Sympt vs. No-N-Sympt (Dominant model A/G-A/A vs. G/G) OR (95% CI): 5.02, (1.40–17.98), *p* = 0.0043; N-Sympt vs. Controls Dominant model (A/G-A/A vs. G/G) OR (95% CI): 3.82, (1.08–13.57), *p* = 0.019; Sept-Sh vs. No-Sept-Sh Dominant model (G/C-C/C vs. G/G) OR (95% CI): 5.19, (1.12–24.04), *p* = 0.014; Sept-Sh vs. Controls Dominant model (G/C-C/C vs. G/G) OR (95% CI): 3.92, (0.85–18.13), *p* = 0.046.

## Data Availability

All data generated or analyzed during this study are stored in electronic archives that can be supplied on request.

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
