# Peer review of "IL-1 Superfamily Member (IL-1A, IL-1B and IL-18) Genetic Variants Influence Susceptibility and Clinical Course of Mediterranean Spotter Fever"

_biomolecules, 2022, doi:10.3390/biom12121892_

Round 1

Reviewer 1 Report

This manuscript describes the identification/typing of genes with predicted relevance to the immune response associated with Rickettsial infection(s), both mild and severe.  The manuscript is important as a potential gateway to understanding genetic markers that may be associated with the clinical course of Rickettsial infections.

Comments:

1.  Correct italicization of genus/species throughout the manuscript.

2.  L50:  PAMPS - Pathogen Associated Molecular Patterns.

3. Change "no canonical" to "non canonical" throughout the manuscript.

4. Change "rickettsia" to "Rickettsia" throughout the manuscript.

5.  Correct capitalization of Genus/species throughout the manuscript.

6.  Section 2.1:  Describe the ELISA assay and PCR assays used to confirm MSF in the patients (including antigens, controls, primers, etc.).

7.  Section 2.1:  Describe the DNA extraction procedure(s).

8.  L152-153:  Finish the first sentence - what has CART been used for?

9.  Section 3:  Divide the Results into specific sections.

10.  L214-216:  How many patients were analyzed - include in the Results (not just in the Tables).

11.  L248-258:  The species of Rickettsia should be mentioned/described in the section.  The immune response can be quite different for selected species of Rickettsia.

12.  L275-278:  Revise the sentence structure (it is difficult to read/interpret as written).

13.  L284-301:  Consider moving this section to the Introduction.  It is a quite lengthy description concerning decision trees and would be better suited to the Introduction of the manuscript.

14.  The Discussion is lacking hypotheses surrounding the mechanistic value/potential of the identified SNPs.  Adding this to the Discussion will greatly improve the Discussion and incorporate relevance into the main findings of the manuscript.  

Author Response

Answers to Referee 1 criticisms and suggestions

This manuscript describes the identification/typing of genes with predicted relevance to the immune response associated with Rickettsial infection(s), both mild and severe.  The manuscript is important as a potential gateway to understanding genetic markers that may be associated with the clinical course of Rickettsial infections.

We are grateful to referee for his positive comment. We have followed his helpful criticisms and suggestions that substantially contributed to increase quality of our manuscript. All changes were marked in red.

Comments:

  1. Correct italicization of genus/species throughout the manuscript.

Thank you for the advice. We have checked manuscript using Italics for all Genus/specie names.

  1. L50:  PAMPS - Pathogen Associated Molecular Patterns.

Sentence has been modified according to referee suggestion

  1. Change "no canonical" to "non canonical" throughout the manuscript.

Done. Thanks for the advice

  1. Change "rickettsia" to "Rickettsia" throughout the manuscript.

Done. We have checked manuscript using Uppercase for Rickettsia Genus, and lowercase for species.

  1. Correct capitalization of Genus/species throughout the manuscript.

Done

  1. Section 2.1:  Describe the ELISA assay and PCR assays used to confirm MSF in the patients (including antigens, controls, primers, etc.).

According to referee suggestion materials and methods were completed with a new section as follows:

Patient serological tests have been performed at admission, during the acute phase, and in the convalescent phase, while molecular tests have been performed on admission only. Rickettsia conorii VIRCLIA® IgM monotest and Rickettsia conorii VIRCLIA® IgG indirect chemioluminescent (CLIA) monotests (Vircell, Granada, Spain) were used for serological tests. VIRCLIA IgM or IgG monotest use the monotest format (a device including Calibrator, Diluent Conjugate and Substrate Chemiluminescence immunoassay on an automatic VIRCLIA® LOTUS SYSTEM. A commercial real time polymerase chain reaction (RT-PCR) Kit was used for qualitative detection of Rickettsia conorii genome (Rickettsia conorii Real-TM, Sacace Biotechnologies s.r.l., Como, Italy). Briefly Rickettsia conorii DNA extracted from the specimens (see below), was amplified using Real-Time amplification and detected by fluorescent reporter dyes linked to hydrolysis probes specific for Rickettsia conorii DNA and Internal Control (IC).

  1. Section 2.1:  Describe the DNA extraction procedure(s).

Following referee suggestion a description of DNA extraction methodology have been inserted in materials and methods as separated section.

  1. L152-153:  Finish the first sentence - what has CART been used for?

We apologize for the typo error . Missed sentence was replaced as follow: Classification and Regression Trees (CART) implementation of a decision tree has been used to check the possibility to build a computer driven decision process that might be useful in discriminating MSF affected or not affected subjects starting from the genetic background variations.

  1. Section 3:  Divide the Results into specific sections.

Following referee suggestion “Results” section was divided into specific subsections

  1. L214-216:  How many patients were analyzed - include in the Results (not just in the Tables).

According to Reviewer suggestion number of subjects are now mentioned in the text and not only in the tables

  1. L248-258:  The species of Rickettsia should be mentioned/described in the section.  The immune response can be quite different for selected species of Rickettsia.

We agree with referee suggestion. Actually Rickettsia species responsible for mild infection or severe induces different cytokine production signature. So the species of Rickettsia studied in papers cited in references were mentioned throughout the manuscript.

  1. L275-278:  Revise the sentence structure (it is difficult to read/interpret as written).

We thank referee for the advice (a part of the original sentence was missed in the manuscript uploaded). Sentence was reshaped at the original  version and we hope that now is readable.

  1. L284-301:  Consider moving this section to the Introduction.  It is a quite lengthy description concerning decision trees and would be better suited to the Introduction of the manuscript.

Description concerning decision tree approach was moved to the “introduction” according to referee suggestion.

  1. The Discussion is lacking hypotheses surrounding the mechanistic value/potential of the identified SNPs.  Adding this to the Discussion will greatly improve the Discussion and incorporate relevance into the main findings of the manuscript.  

We are indebted with referee for this suggestion. A paragraph on hypotheses on the role that the presence of  IL-1 and IL-18 SNPs might play in MSF, and the possibility of SNP typing driven therapeutic approach, have been added to discussion

Other changes of the paper have been performed according with the suggestions of other referees

Reviewer 2 Report

This is a very interesting work focusing on the association of single nucleotide polymorphisms (SNPs) of pro-inflammatory genes with the severity of Mediterranean spotted fever (MSF). The study included a total of 366 subjects with 160 patients who diagnosed as MSF. Several functional and common SNPs of IL-1 clusters were analyzed by competitive allele specific PCR. The major conclusions of this work were IL-1A rs1800587TT genotype was significantly increased in patients alone. IL-1 gene cluster SNP genotypes and IL-18 rs187238G positive genotypes were associated to risk of severe complications. The research topic is significant and interesting. However, the enthusiasms were significantly dampened by several major weaknesses of the studies. The major weakness is that the details on diagnosis for MSF were not mentioned at all. Is remains unclear whether these SNPs of cytokine genes were induced by infection or associating with host susceptibility. There were also several incorrect statements as listed below, interfering better understanding the studies. In addition, extensive editing of English language and style are required due to quite a few incorrect words or sentences.

11)      It remained unknow on how the patients were diagnosed with MSF. The details on how many patients were diagnosed based on PCR or serology remained unclear. These are required for accurately understanding the data and drawing the appropriate conclusion.

22)      It remained unknow when the blood samples were collected. This is another important piece of information for better understanding the results considering the course of the disease. Were these sample collected after the recovery or at the acute stage of infection?

33)      Line 70, “Initial activation of canonical inflammasome results in the maturation of …. TFN-alpha… .” It is well-known that TNF-alpha is a pro-inflammatory cytokine independent of inflammasome activation.

44)      Line 259, “Relevant SNPs that might modify activation of inflammasome” is confusing. The IL-1 clusters, except IL-1alpha, including IL-1beta and IL-18, were downstream event after inflammasome activation. How would these cytokines modify the activation of inflammasome? Affecting signal 1 or signal 2?

55)      Line 73, “IL-1a is …released by the no canonical inflammasome pathway….” In this manuscript, IL-1a and IL-1A were used but not know whether they are interchangeable. It is confusing whether these two referred to “IL-1alpha”? If it is, IL-1alpha can be released in a way independent of inflammasome.

66)      “No canonical” should be “Non-canonical”.

77)      Table 2, TIRAP, “Reduced receptor signal transduction…” Why reduced?

88)      Table 3, Did “Nr” represent "number"? It will be helpful to have footnote displaying all the abbreviation in each table.

99)      TNF-alpha has been published in previous studies with very similar experimental setting, as indicated in one of the references.

110)   Figure 1 is very confusing. Not sure what exactly means.

111)   Line 206 “Lesser”?

112)   Table 6 and Table 7 employed different indicators for severe MSF. Table 6 included “Sept” and neurological systems while Table 7 only focused on ARDS. No underlying justification was made. Would these different SNPs be indicative for different severe symptom?

113)   Line 293 “ labeled data dataset” ?

Author Response

Answers to Referee 2 criticisms and suggestions 

This is a very interesting work focusing on the association of single nucleotide polymorphisms (SNPs) of pro-inflammatory genes with the severity of Mediterranean spotted fever (MSF). The study included a total of 366 subjects with 160 patients who diagnosed as MSF. Several functional and common SNPs of IL-1 clusters were analyzed by competitive allele specific PCR. The major conclusions of this work were IL-1A rs1800587TT genotype was significantly increased in patients alone. IL-1 gene cluster SNP genotypes and IL-18 rs187238G positive genotypes were associated to risk of severe complications. The research topic is significant and interesting. However, the enthusiasms were significantly dampened by several major weaknesses of the studies. The major weakness is that the details on diagnosis for MSF were not mentioned at all. Is remains unclear whether these SNPs of cytokine genes were induced by infection or associating with host susceptibility. There were also several incorrect statements as listed below, interfering better understanding the studies. In addition, extensive editing of English language and style are required due to quite a few incorrect words or sentences.

Authors are grateful to reviewer for positive comments and helpful criticisms and suggestions that were all addressed, paper grammar and style of the manuscript had been revised. We feel that reviewer criticisms and suggestions substantially contributed to increase quality of our manuscript. All changes were marked in red

11)      It remained unknow on how the patients were diagnosed with MSF. The details on how many patients were diagnosed based on PCR or serology remained unclear. These are required for accurately understanding the data and drawing the appropriate conclusion.

We agree with referee comments. According to suggestions a more detailed description of times and methodological schedules applied to R. Conorii infection diagnosis was inserted in “material and methods” section 

22)      It remained unknow when the blood samples were collected. This is another important piece of information for better understanding the results considering the course of the disease. Were these sample collected after the recovery or at the acute stage of infection?

We are grateful to referee for advice. As above reported we have inserted in the in “material and methods” section a more detailed description of timing of laboratory R. Conorii infection diagnosis. Sample for human genoma SNP typing was obtained, as now reported in M.M., at the first day of hospital admission. 

33)      Line 70, “Initial activation of canonical inflammasome results in the maturation of …. TFN-alpha… .” It is well-known that TNF-alpha is a pro-inflammatory cytokine independent of inflammasome activation.

We agree with referee comment. Accordingly (see also answer to query 99) we have eliminated data on TNFA typing from revised version of the manuscript. Actually, present data on TNFA rs1800629 even if confirm our previously published observation, do not add substantial novelty

44)      Line 259, “Relevant SNPs that might modify activation of inflammasome” is confusing. The IL-1 clusters, except IL-1alpha, including IL-1beta and IL-18, were downstream event after inflammasome activation. How would these cytokines modify the activation of inflammasome? Affecting signal 1 or signal 2?

We apologize for inducing this misunderstanding. We have changed the sentence to avoid all possible ambiguity as follows: “In this view we have evaluated some functionally relevant SNPs that might modify activation of inflammasome (in particular TIRAP rs8177374) and related cytokine production (rs1800587, rs16944 and rs1143634 of IL-1 gene cluster and IL-18 rs187238).”

55)      Line 73, “IL-1a is …released by the no canonical inflammasome pathway….” In this manuscript, IL-1a and IL-1A were used but not know whether they are interchangeable. It is confusing whether these two referred to “IL-1alpha”? If it is, IL-1alpha can be released in a way independent of inflammasome.

We agree with referee comment as IL-1alpha can be released in a way independent of inflammasome, and this is now stated in the introduction. Revised version of the manuscript indicate with “IL-1A” the gene and with “IL-1α” the protein coded by the gene. 

66)      “No canonical” should be “Non-canonical”.

Done, thanks for the advice

77)      Table 2, TIRAP, “Reduced receptor signal transduction…” Why reduced?

According to referee request a further reference was added in the table. The structural explanation of functional effect of SNP minor allele was inserted in the discussion as follow: “Cristal structure studies of TIRAP gene have demonstrated that S180 residue is located in a surface-exposed cavity and that substitution by leucine is likely to cause steric occlusion of the cavity. MyD88 may bind in a different configuration to MAL S180L variant, which interfere with the formation of the receptor ternary complexes required to initiate signaling”.

88)      Table 3, Did “Nr” represent "number"? It will be helpful to have footnote displaying all the abbreviation in each table.

According to referee suggestion, we have checked that all the abbreviation were displayed in all table headers or footnotes. Nr was substitute with N. meaning “number” .

99)      TNF-alpha has been published in previous studies with very similar experimental setting, as indicated in one of the references.

We agree with referee criticism. Accordingly, as data on TNF-alpha reported in this paper do not add substantial novelty  respect to our previously published data we have eliminated data on TNFA typing from revised version of the manuscript

110)   Figure 1 is very confusing. Not sure what exactly means.

According with referee observation, Figure and figure caption was implemented to clarify the mean.

111)   Line 206 “Lesser”?

Sentence has been reshaped

112)   Table 6 and Table 7 employed different indicators for severe MSF. Table 6 included “Sept” and neurological systems while Table 7 only focused on ARDS. No underlying justification was made. Would these different SNPs be indicative for different severe symptom?

Table 6 and 7 were extracted from supplementary tables S1 to S5. In each supplementary table associations of all SNPs typed with a single severe symptom were analyzed, and the old table 6 and 7 summarize the statistically significant results reported in the supplementary materials. To facilitate the reading, in the revised version of the manuscript, we have detailed in the text what each supplementary table contain and renumbered table 6 as 7 and table 7 as 6 because table S1 refers to ARDS and tables S2 to S5 to the other symptoms. Data seem to suggest that IL-18 SNP is a susceptibility factor for ARDS whereas different SNPs of IL-1 gene cluster  might be susceptibility factor for the other symptoms and in particular for sepsis and subsequent complications (septic shock and coma). We have clearly stated that we consider with caution these preliminary evidences that should be confirmed in larger sets of patients with severe symptoms, however is intriguing that a genetically determined higher production of inflammatory cytokines seem to be associated with a more severe MSF course.

113)   Line 293 “ labeled data dataset” ?

A general description concerning decision tree approach was moved to the “introduction” section reshaping and simplifying “technicistic” sentences. According with figure 1 restyling  decision tree section of the results and corresponding discussion paragraphs are rewritten 

Other changes of the paper have been performed according with the suggestions of other referees